# A Symmetrized Dot Pattern Extraction Method Based on Frobenius and Nuclear Hybrid Norm Penalized Robust Principal Component Analysis and Decomposition and Reconstruction

**DOI:** 10.3390/s23208509

**Published:** 2023-10-17

**Authors:** Lijing Wang, Shichun Wei, Tao Xi, Hongjiang Li

**Affiliations:** 1School Control and Mechanical Engineering, Tianjin Chengjian University, Tianjin 300384, China; xiwang8668@126.com (L.W.); 18822593936@163.com (S.W.); 15222521538@163.com (H.L.); 2School of Mechanical Engineering, Tiangong University, Tianjin 300387, China

**Keywords:** rolling bearing, fault feature extraction, SDP, FNHN-RPCA, decomposition and reconstruction, DPR/KLdiv

## Abstract

Due to their symmetrized dot pattern, rolling bearings are more susceptible to noise than time–frequency characteristics. Therefore, this article proposes a symmetrized dot pattern extraction method based on the Frobenius and nuclear hybrid norm penalized robust principal component analysis (FNHN-RPCA) as well as decomposition and reconstruction. This method focuses on denoising the vibration signal before calculating the symmetric dot pattern. Firstly, the FNHN-RPCA is used to remove the non-correlation between variables to realize the separation of feature information and interference noise. After, the residual interference noise, irrelevant information, and fault features in the separated signal are clearly located in different frequency bands. Then, the ensemble empirical mode decomposition is applied to decompose this information into different intrinsic mode function components, and the improved DPR/KLdiv criterion is used to select components containing fault features for reconstruction. In addition, the symmetrized dot pattern is used to visualize the reconstructed signal. Finally, method validation and comparative analysis are conducted on the CWRU datasets and experimental bench data, respectively. The results show that the improved criteria can accurately complete the screening task, and the proposed method can effectively reduce the impact of strong noise interference on SDPs.

## 1. Introduction

As a key component of rotating mechanical equipment, such as engines [1], wind turbines [2], electric machines [3], and so on, a rolling bearing has long-term exposure to complex and high-intensity work environments, and various types of faults may occur. Due to the different working conditions and the strong interference from other sources, it is difficult to distinguish the two-dimensional image features reflected by the vibration signal [4,5]. Therefore, extracting accurate and stable two-dimensional fault features is of great significance for accurately identifying the health status of bearings.

A symmetrized dot pattern (SDP) [6] can achieve a two-dimensional visualization of one-dimensional vibration signals. This method converts one-dimensional vibration signals into a snowflake graph in a two-dimensional polar coordinate system using time intervals, angle amplification coefficients, and symmetrical distributions. The shapes of each pattern arm in the figure can well reflect the differences between vibration signals [7,8]. However, the collected vibration signals of rolling bearings are usually non-linear and non-stationary pollution signals, and directly processing these data with an SDP can result in a high similarity in SDP feature images, requiring feature enhancement of the signals before converting them into SDP images. Robust principal component analysis (RPCA), as a matrix low-rank sparse decomposition model, can achieve the separation of feature information and interference noise by removing non-correlation between variables, thereby improving the proportion of feature information in the data [9]. In previous studies, RPCA has been widely applied in the field of computer vision [10,11], but it is still in the exploratory stage in the field of fault diagnosis. In recent years, Mao et al. used RPCA to extract the characteristics of a low-dimensional submanifold structure from a signal trajectory matrix to suppress background noise in the bearing vibrational signal [12]. Based on the sensitivity of RPCA to severely damaged data, Tang et al. extracted a fault feature in the bearing vibration signal under a low-speed condition from extreme background noise [13]. Although the above studies have achieved some good results, the non-convex constraint problem of the RPCA method cannot be ignored. Regarding this issue, Shang et al. proposed the FNHN penalty term and established a model of bilinear factor matrix norm minimization, where the norm of each bilinear factor matrix is the convex form [14]. On this basis, Yu et al. used the FNHN-RPCA solver to accurately extract the rolling bearing transient pulse features contaminated by noise, which brought forward a new perspective for FNHN-RPCA in the field of fault diagnosis [15].

The time–frequency analysis (TFA) method has a remarkable ability to extract the hidden features of the signal [16,17]. At present, many advanced TFA methods have been proposed, such as Empirical Wavelet Transform (EWT) [18], Variational Mode Decomposition (VMD) [19], Ensemble Empirical Mode Decomposition (EEMD) [20], etc. Compared with other time–frequency analysis methods, EEMD reduces the number of optimal components and the quadratic penalty factor in VMD, the basis function in SWT, and the complex calculations in other time–frequency analysis methods, and it can adaptively decompose complex signals into a finite number of intrinsic mode functions (IMFs). However, all EEMD-based methods face the following issue: does the IMF contain fault feature information?

For ideal vibration signals with weak or no noise, only entropy features are needed to filter out weak noise IMFs and redundant information components [21]. An IMF can also be selected using any combination of statistical features such as variance contribution [22], correlation coefficient [23], and mean period [24] or several default components to reconstruct the fault feature signal can be directly selected [25,26]. When the vibration signal is strongly interfered with by noise, using the EEMD method will cause the noise to be distributed in various IMFs without quantification. At the same time, the existence of noise also affects the selection of a fault feature component, which makes the decomposition reconstruction unable to extract the ideal fault feature signal. In order to extract the fault characteristics of the friction impact of different strengths, Alexander et al. proposed an IMF selection criterion combining the degree-of-presence ratio (DPR), the Kullback–Leibler divergence-based similarity measure (KLdiv) [27], and an adaptive selection threshold method [28]. The proportion of fault information is represented by calculating the ratio of the DPR and KL divergence of each component. Inspired by this, this article will improve the DPR/KLdiv criterion for selecting IMF components under strong noise interference.

In this paper, in order to eliminate the influence of strong noise interference on an SDP pattern, firstly, the one-dimensional vibration signal containing noise is converted into a fault information matrix based on phase space reconstruction, and FNHN-RPCA is introduced to decompose it into a low-rank information matrix (LRM) and a sparse background noise matrix. Then, the LRM is rearranged into a one-dimensional feature enhancement signal. Due to the strong separation ability of the FNHN-RPCA solver, most of the noise, irrelevant information, and fault features will be in different frequency bands. Next, the use of EEMD to decompose the information contained in the enhanced feature signal into different IMF components and the improved DPR/KLdiv criterion are used to select the IMF containing fault features to reconstruct the fault feature signal. Among them, the DPR/KLdiv selection criterion is improved by redefining the DPR part and adaptive selection threshold in the criterion. Finally, the SDP method is used to calculate the SDP features of the fault feature signal. Thanks to the strong separation ability of the FNHN-RPCA and the accuracy of the improved DPR/KLdiv criterion, the SDP extraction method proposed in this paper can effectively reduce the impact of strong noise interference on an SDP, significantly reduce the similarity between bearing SDP features with different health levels, and increase the stability of an SDP.

The rest of this paper is organized as follows. Section 2 introduces the FNHN-RPCA theory, the DPR/KLdiv selection criteria, and the principles of an SDP. The proposed methods are also described, including the improvements in DPR and adaptive thresholds, the use of FNHN-RPCA, EEMD, improved DPR/KLdiv, and SDP methods for an SDP extraction. Section 3 conducts experimental verification and result analysis. Section 4 summarizes the research of this article.

## 2. Related Theories and Proposed Methods

### 2.1. Frobenius and Nuclear Hybrid Norm Penalized Robust Principal Component Analysis

Let the one-dimensional vibration signal be *x* = [*x*_1_,*x*_2_,...,*x*_n_], and convert *x* into a high-dimensional information matrix using phase space reconstruction. This can be described as follows:(1)$$X=\left[\begin{array}{cccccc} x_{1} & x_{2} & \cdots & x_{\tau } & \cdots & x_{n_{2}}\\ x_{\tau +1} & x_{\tau +2} & \cdots & x_{2\tau } & \cdots & x_{2n_{2}}\\ \vdots & \vdots & \cdots & \vdots & \cdots & \vdots \\ x_{(n_{1}-1)\tau +1} & x_{(n_{1}-1)\tau +2} & \cdots & x_{(n_{1}-1)\tau +\tau } & \ddots & x_{(n_{1}-1)\tau +n_{2}} \end{array}\right]$$
where *n*_1_ is the embedding dimension and *τ* is the delay time ((*n*_1_ − 1)*τ* + *n*_2_ = *n*) [29]. In this paper, the specific principles of matrix construction will not be explained; references can be found in reference [15]. Although there are differences in the construction forms in different references, the fundamental idea of construction is the same.

Using RPCA, the information matrix *X* can be decomposed into a low-rank approximate information matrix and a sparse background noise matrix. The RPCA model is expressed as [9]:(2)$$\underset{L,S}{\min}\lambda rank_{}(L)+||S|{|}_{l_{0}}{,}_{}s.t._{}L+S=X$$
where $||S|{|}_{l_{0}}$ is the *l*_0_-norm of the sparse matrix *S* and *λ* is the regularization parameter used to balance the weight between the low-rank matrix *L* and the sparse matrix *S*.

Due to the non-convexity of the rank function and the *l*_0_-norm in Equation (2) in optimization, it belongs to the NP-hard problem. Therefore, convex or non-convex proxies are usually used to replace the above two terms in Equation (2), and the above problem can be represented as follows [30,31]:(3)$$\underset{L,S}{\min}\lambda ||L|{|}_{S_{q}}^{q}+||S|{|}_{l_{p}}^{p}{,}_{}s.t._{}P_{\Omega }(L+S)=P_{\Omega }(X)$$
where p,q∈[0,2], $||L|{|}_{S_{q}}$, and $||S|{|}_{l_{p}}$ are considered regularized and loss terms, respectively. *P*_Ω_ is an orthogonal projection on a linear subspace that supports a matrix on Ω.

#### 2.1.1. The Principle and Framework of FNHN-RPCA

FNHN-RPCA relaxes the problem of classical RPCA non-convex constraints with FNHN penalty items, making RPCA problems straightforward to handle and extend [14]. The main principles of FNHN-RPCA are as follows.

Any matrix $X\in R^{n_{1}\times n_{2}}$ that satisfies rank *r* ≤ *d* can be expressed as two-factor matrices $U\in R^{n_{1}\times d}$ and $V\in R^{n_{2}\times d}$, $X=UV^{T}$. Then, define the FNHN penalty of the matrix *X* as:(4)$$||X|{|}_{F-N}=\underset{U,V:X=UV^{T}}{\min}{\left[(||U|{|}_{F}^{2}+2||V||*)/3\right]}^{3/2}$$

This proves $||X|{|}_{F-N}$ is a $||X|{|}_{S_{2/3}}$ (Schatten-2/3 quasi-norm), where ||⋅||∗ is the nuclear norm.

For any matrix, $X=UV^{T}\in R^{n_{1}\times n_{2}}$, the following inequality is satisfied:(5)$$\frac{1}{3}(||U|{|}_{F}^{2}+2||V||*)\geq ||X|{|}_{S_{2/3}}^{2/3}$$

Therefore, matrix $X\in R^{n_{1}\times n_{2}}$ satisfies the following equation:(6)$$\begin{array}{cc} ||X|{|}_{F-N} & =\underset{U\in R^{n_{1}\times d},V\in R^{n_{2}\times d}:X=UV^{T}}{\min}{\left(\frac{\left(||U|{|}_{F}^{2}+2||V||*\right)}{3}\right)}^{3/2}\\ & =\underset{U,V:X=UV^{T}}{\min}||U|{|}_{F}^{}||V||* \end{array}$$

The boundary of the FNHN penalty is:(7)$$||X||*\leq ||X|{|}_{F-N}\leq rank(X)||X||*$$

This suggests that the FNHN penalty can be seen as an approximation of the nuclear norm.

Let $P_{\Omega^{C}}(X)=0$ and $S_{\Omega^{C}}\in （-\infty ，+\infty ）$; then, $P_{\Omega^{C}}(L+S)=P_{\Omega^{C}}(X)$, where Ω*^C^* is the complement of Ω. The constraint on Equation (3) operating *P*_Ω_ using orthogonal projection is replaced with *L+S=X*. Based on the super-Laplace prior to sparse components, low-rank approximate information is defined by Equation (5), where $||L|{|}_{F-N}^{2/3}$ replaces $||L|{|}_{S_{q}}^{q}$ in Equation (3). Then, the FNHN-RPCA model is expressed as follows:(8)$$\underset{U,V,L,S}{\min}\frac{\lambda }{3}(||U|{|}_{F}^{2}+2||V||*)+||P_{\Omega }(S)|{|}_{l_{2/3}}^{2/3},s.t.UV^{T}=L,L+S=X$$

#### 2.1.2. Optimization Algorithm for FNHN-RPCA

An auxiliary variable $\hat{V}$ is introduced into Equation (8) for ease of solving. Then, the model can be reformed as follows:(9)$$\begin{array}{l} \underset{U,V,L,S,\hat{V}}{\min}\frac{\lambda }{3}(||U|{|}_{F}^{2}+2||\hat{V}||*)+||P_{\Omega }(S)|{|}_{l_{2/3}}^{2/3},\\ s.t.\hat{V}=V,UV^{T}=L,L+S=X \end{array}$$

The augmented Lagrangian form of Equation (9) is as follows:(10)$$\begin{array}{l} L_{\mu }(U,V,L,S,\hat{V},Y_{1},Y_{2},Y_{3})=\frac{\lambda }{3}(||U|{|}_{F}^{2}+2||\hat{V}||*)+||P_{\Omega }(S)|{|}_{l_{2/3}}^{2/3}\\ +\langle Y_{1},\hat{V}-V\rangle +\langle Y_{2},UV^{T}-L\rangle +\langle Y_{3},L+S-X\rangle \\ +\frac{\mu }{2}(||\hat{V}-V|{|}_{F}^{2}+||UV^{T}-L|{|}_{F}^{2}+||L+S-X|{|}_{F}^{2}) \end{array}$$
where $\langle \cdot ,\cdot \rangle$ represents the inner product operator between elements in two matrices. $Y_{1}\in R^{n_{1}\times d}$, $Y_{2}\in R^{n_{2}\times d}$, and $Y_{3}\in R^{n_{1}\times n_{2}}$ are the matrices of the Lagrange multiplier. μ>0 indicates a penalty factor. Then, introduce the ADMM algorithm to solve Equation (10), as shown in Figure 1. The main steps of solving are as follows:

Step 1: Update parameters *U_k_*_+1_ and *V_k_*_+1_. Update *U_k_*_+1_ and *V_k_*_+1_ by solving the following two optimization problems, respectively:(11)$$\underset{U}{\min}\frac{\lambda }{3}||U|{|}_{F}^{2}+\frac{\mu_{k}}{2}||UV_{k}^{T}-L_{k}+\mu_{k}^{-1}Y_{2}^{k}|{|}_{F}^{2}$$
(12)$$\underset{V}{\min}||\hat{V}-V+\mu_{k}^{-1}Y_{1}^{k}|{|}_{F}^{2}+||U_{k+1}V^{T}-L_{k}+\mu_{k}^{-1}Y_{2}^{k}|{|}_{F}^{2}$$

The optimal solutions for these two optimization problems are as follows:(13)$$U_{k+1}=\mu_{k}P_{k}V_{k}{\left(\frac{2\lambda }{3}I_{d}+\mu_{k}V_{k}^{T}V_{k}\right)}^{-1}$$
(14)$$V_{k+1}=\left({\hat{V}}_{k}+\mu_{k}{}^{-1}Y_{1}^{k}+H_{k}^{T}U_{k+1}\right){\left(I_{d}+U_{k+1}^{T}U_{k+1}\right)}^{-1}$$
where $H_{k}=L_{k}-\mu_{k}^{-1}Y_{2}^{k}$ and *I_d_* represents an identity matrix of size *d*×*d*.

Step 2: Update the parameter ${\hat{V}}_{k+1}$. With other variables constant, update ${\hat{V}}_{k+1}$ by solving the following optimization problem:(15)$$\underset{\hat{V}}{\min}\frac{2\lambda }{3}||\hat{V}||*+\frac{\mu_{k}}{2}||\hat{V}-V_{k+1}+Y_{1}^{k}/\mu_{k}|{|}_{F}^{2}$$

Step 3: Update the parameter *L_k_*_+1_. Update *L_k_*_+1_ by solving the following optimization problem:(16)$$\underset{L}{\min}||U_{k+1}V_{k+1}^{T}-L+\mu_{k}^{-1}Y_{2}^{k}|{|}_{F}^{2}+||L+S_{k}-X+\mu_{k}^{-1}Y_{3}^{k}|{|}_{F}^{2}$$

Its optimal solution is as follows:(17)$$L_{k+1}=\frac{1}{2}(U_{k+1}V_{k+1}^{T}+\mu_{k}^{-1}Y_{2}^{k}-S_{k}+X-\mu_{k}^{-1}Y_{3}^{k})$$

Step 4: Update the parameter *S_k_*_+1_. To update *S_k_*_+1_, solve the optimization problem below Equation (17) while fixing the other variables.
(18)$$\underset{S}{\min}||P_{\Omega }(S)|{|}_{l_{2/3}}^{2/3}+\frac{\mu_{k}}{2}||S+L_{k+1}-X+\mu_{k}^{-1}Y_{3}^{k}|{|}_{F}^{2}$$

The above problem can be effectively solved by introducing a two-thirds threshold operator, represented as:(19)$$S_{k+1}=P_{\Omega }(\tau_{2/\mu_{k}}(X-L_{k+1}-\mu_{k}^{-1}Y_{3}^{k})+P_{\frac{1}{\Omega }}(X-L_{k+1}-\mu_{k}^{-1}Y_{3}^{k})$$

### 2.2. The DPR/KLdiv Selection Criteria

After using EEMD to decompose the signal, multiple IMF components will be generated, and it is necessary to select the components related to fault features accurately. This article refers to the selection criteria proposed in reference [28] that combine the degree-of-presence ratio (DPR) and Kullback–Leibler divergence-based similarity measure (KLdiv). DPR can detect the presence and power of fault frequencies and their harmonic frequencies in the power spectrum of the IMFs. KLdiv can discover the correlation between the original signal and the selected components in hidden structures. The calculation steps for DPR are as follows:

Step 1: Assuming *x*(t) is an arbitrary IMF component, the power spectrum of the IMF can be calculated using Hilbert transform and Fourier transform.

Step 2: Construct a Gaussian mixture model (GMM) window, and the GMM window function $G_{window}(k,\delta )$ is calculated as follows [27,32]:(20)$$G_{window}(k,\delta )=\left\{\begin{array}{l} \sum_{i=1}^{n}\exp\left(-\frac{1}{2}{\left(\delta \frac{\left(k-FOI_{n}\right)}{\left(N_{rfreq}/2\right)}\right)}^{2}\right)\\ 0,\mathrm{otherwise} \end{array}\right.,FOI_{n}-f_{range}\leq k\leq FOI_{n}+f_{range}$$
where $\delta =(N_{rfreq}/N_{wfreq})\sqrt{2\ln m}(m=1)$ is a Gaussian random variable inversely proportional to the standard deviation and *FOI_n_* is the *n_th_* harmonic of the fault frequency (*n* = 3). $N_{wfreq}=((2\times 2/100)\times FOI)/f_{resolution}$ is the size of the frequency bin around the fault frequency. $N_{rfreq}=2f_{range}/f_{resolution}$ is the frequency bins within the Gaussian mixing window range ($f_{range}=FOI/3$, $f_{resolution}=1$).

Step 3: The fault frequency components of the power spectrum are calculated by multiplying the GMM window, and the residual frequency component is calculated by subtracting the fault frequency component from the IMF power spectrum. The calculation process for the fault frequency and residual frequency components is shown in Figure 2.

Step 4: The parameter DPR is calculated using the ratio of the fault frequency component to the residual frequency component, and the calculation formula for *DPR* is as follows:(21)$$DPR=10\cdot \log\left(\sum_{n=1}^{3}\left\{\sum_{j=1}^{N_{wfreq}}C_{n,j}^{2}/\sum_{j=1}^{N_{rfreq}}R_{n,j}^{2}\right\}+10\right)(dB)$$
where *C_n_*_, *j*_ and *R_n_*_, *j*_ are the sizes of the *j_th_* frequency bin near the *n_th_* harmonic of the fault frequency component and the residual frequency component.

The KLdiv section is calculated using the probability density functions of the original signal and each component [33], and its calculation formula is as follows:(22)$$KLdiv=\sum_{z\in Z}PDF(z(t))\ln\left(\frac{PDF(z(t))}{PDF(IMF_{k}(t))}\right)$$
where *PDF*(*z*(*t*)) and *PDF*(*IMF_k_*(*t*)), respectively, represent the probability density function of the original vibration signal and the *k_th_* IMF. Therefore, the evaluation index *Obj_k_* of the *k_th_* IMF is expressed as:(23)$$Obj_{k}=DPR_{k}/KLdiv_{k}$$

If the value of normalized *Obj* calculated with each component is greater than the adaptive threshold adTH, then this component is selected as the fault information component. The threshold *adTH* is defined as:(24)$$adTH=\mathrm{var}\sqrt{2\log N}$$
where var represents the variance of the normalized *Obj* value and *N* is the total number of IMF components.

### 2.3. SDP Theory

A symmetrized dot pattern (SDP) describes one-dimensional vibration signals in polar coordinates system and reflects vibration information through images. Assuming that the time–domain vibration signal is *y* = {*y*_1_, *y*_2_,..., *y*_n_}, any point *i* can be converted to point *P*(*r*(*i*), *θ*(*i*), *ϕ*(*i*)) in the polar coordinate system using the following reference [6]. The principle is shown in Figure 3.
(25)$$r(i)=\frac{y_{i}-y_{\min}}{y_{\max}-y_{\min}}$$
(26)$$\theta (i)=\theta +\frac{y_{i+H}-y_{\min}}{y_{\max}-y_{\min}}\zeta$$
(27)$$\varphi (i)=\theta -\frac{y_{i+H}-y_{\min}}{y_{\max}-y_{\min}}\zeta$$
where *r*(*i*) is the radius of the polar coordinate system, *θ*(*i*) is the counterclockwise rotation angle of the polar coordinate along the initial line, and *ϕ*(*i*) is the clockwise rotation angle of the polar coordinate along the initial line. *y*_max_ and *y*_min_ represent the maximum and minimum values in the data and *H* is the time interval. *θ* is the rotation angle of the symmetry plane (*θ* = 60°). *ζ* is the amplification factor of angle (*ζ* ≤ *θ*). Combining Equations (24)–(26) and Figure 3 shows that the selection of *H* and *ζ* directly affects the SDP image characteristics.

### 2.4. The Proposed Method

#### 2.4.1. Improvements to the DPR/KLdiv Criterion

In order to adapt the DPR/KLdiv criterion to the selection of components under strong noise conditions, this article has made improvements to this criterion.

(1) Under substantial noise interference, the ratio of the fault frequency component to the residual frequency component in Equation (21) will be small, resulting in a lower and closer differentiation of the DPR calculated using all components, as shown in Figure 4. Therefore, this paper replaces the envelope power spectrum required for calculating the DPR with the envelope spectrum and redefines the *DPR* as *DPR*′:(28)$$DPR^{\prime }=\log\left(\sum_{n=1}^{3}\left\{\sum_{j=1}^{N_{wfreq}}C_{n,j}^{2}/\sum_{j=1}^{N_{rfreq}}R_{n,j}^{2}\right\}*10\right)$$

(2) Meanwhile, due to noise interference, the adaptive threshold *adTH* calculated using Equation (24) will result in a problem close to the evaluation index Obj value of some components. Therefore, this article redefines the adaptive threshold as:(29)$$aw=\frac{mean+\mathrm{var}\sqrt{2\log N}}{2}$$
where *mean* represents the mean of the normalized objective function value, var represents the variance, and *N* is the total number of IMF.

#### 2.4.2. SDP Pattern Extraction Model Based on FNHN-RPCA and Decomposition and Reconstruction

First, we use phase space reconstruction, the FNHN-RPCA algorithm, and restoration operation to enhance the features of the collected vibration signals. Then, the enhanced signal is denoised using EEMD and the improved DPR/KLdiv criterion, and the feature extraction process is shown in Figure 5. The steps are as follows:

Step 1: Construct the collected one-dimensional vibration signal *x*_1_(*t*) into an information matrix *X*_1_.

Step 2: Using FNHN-RPCA to decompose *X*_1_ into a low-rank approximate information matrix *L*_1_ and a sparse background noise matrix *S*_1_, then restore *L*_1_ to feature enhanced signal *x*_2_(*t*).

Step 3: Decompose the enhanced signal *x*_2_(*t*) into *n* IMFs using EEMD. 

Step 4: Use the improved DPR/KLdiv criterion to filter these components and reconstruct the selected components into fault feature signals *x*_3_(*t*).

Step 5: Convert *x*_3_(*t*) into an SDP image.

## 3. Experimental Verification and Results Analysis

This section will perform experimental verification and results analysis using a Case Western Reserve University public dataset and a rolling bearing test-rig dataset, respectively, to validate the accuracy of the improved DPR/KLdiv criterion and to evaluate the performance of the proposed SDP feature extraction method.

### 3.1. Case 1. Western Reserve University Public Dataset

#### 3.1.1. Experimental Instruments and Experimental Data

To verify the effectiveness of the proposed SDP extraction method based on FNHN-RPCA and decomposition reconstruction, this work was conducted on the rolling bearing dataset published by Case Western Reserve University (CWRU), and the experimental platform is shown in Figure 6.

This paper studies the vibration data of the motor drive end bearing under a sampling frequency of 12 kHz and a load of 2 HP (1750 r/min). The health status of bearings is set to 6 categories: normal, inner race fault, outer race fault@12, outer race fault@6, outer race fault@3, and ball fault. Since the vibration data in the dataset are hardly disturbed by noise, this paper incorporates random shocks and Gaussian noise from reference [34] into the dataset.
(30)$$o(t)=\underset{\begin{array}{c} \;\mathrm{Original}\;\\ \;\mathrm{signal} \end{array}}{\underset{︸}{x(t)}}+\underset{\mathrm{Random}\;\mathrm{shocks}\;}{\underset{︸}{\sum_{j}R_{j}\cdot S(t-r_{j})}}+\underset{\begin{array}{c} \;\mathrm{Gaussian}\;\\ \;\mathrm{noise} \end{array}}{\underset{︸}{n(t)}}$$
where random shocks represent occasional impulse or electromagnetic interference introduced during data collection. If $S(t)=e^{-vt}\sin(2\pi ft)$ (*t* > 0), and selecting *R*_1_ = 1.5, *r*_1_ = 0.12, *f* = 4500, *v* = 480, then the data for the six health states of the bearing are shown in Figure 7. The SDP patterns converted from pre- and post-noise data are shown in Figure 8. It can be seen that the added noise significantly changes the curvature, thickness, and dispersion and concentration areas of the pattern arm in the SDP pattern, resulting in a minor differentiation among the SDP pattern of different faults (the parameter used to calculate SDP is set to time interval *H*=2 and amplification factor of angle *ζ* = 30°).

#### 3.1.2. Improve the Validation of the DPR/KLdiv Criteria

To verify the accuracy of the improved DPR/KLdiv criterion, this section will compare the DPR values, Obj values, number of component selections, envelope spectrum, and SDP pattern of the reconstructed signal of the DPR/KLdiv criterion before and after the improvement.

Take the inner race fault noise signal with a damage degree of 7 mil as an example. The waveform and envelope spectrum of the bearing inner race fault signal before noise addition are shown in Figure 9a,b. From the envelope spectrum, the characteristic frequency (*f*_i_) of the inner race fault and related content of its harmonic frequency (2*f*_i_~6*f*_i_) are evident. Figure 9c,d show the waveform and envelope spectrum of the noisy signal. Due to the substantial interference of noise, many irregular and prominent peaks in the envelope spectrum make it challenging to observe prominent fault characteristics.

Firstly, we use FNHN-RPCA to enhance the fault features of the noisy signal in Figure 9c (the phase space reconstruction parameters used in RPCA are set to *n*_1_ = 120 and *τ* = 100) to obtain the enhanced signal. Then, we use EEMD to decompose the enhanced signal into 14 IMF components, and the envelope spectra of each component are shown in Figure 10. The figure shows that the noise is mainly concentrated in the high-frequency component IMF1~3, and the low-frequency component contains almost no content related to the fault characteristic frequency. We use the original DPR/KLdiv criterion to select components that contain fault information and have relatively low noise, and the DPR values, KLdiv values, Obj values, and threshold adTH during the calculation process are listed in Table 1.

Table 1 show that the DPR values of each IMF are close to 10, and at this point, the DPR/KLdiv criterion is entirely determined by the KLdiv value and has lost its significance. This phenomenon is because the ratio of the fault frequency component and the residual frequency component in Equation (21) is small. The picture of component selection drawn from the data in Table 1 is shown in Figure 11a; the DPR/KLdiv criterion selected IMF2~14 for reconstruction, which does not match the results shown in Figure 10. The selected components include noise component IMF2~3 and some low-frequency components unrelated to fault characteristics. We reconstructed the selected components, and the waveform and envelope spectrum of the reconstructed signal are shown in Figure 11b,c. From the waveform (Figure 11b), it can be seen that the signal still contains interference noise. However, thanks to the powerful feature enhancement ability of FNHN-RPCA, the envelope spectrum (Figure 11c) can clearly display the fault characteristics. The SDP feature image of the reconstructed signal is shown in Figure 11d, and there is no significant change in the thickness and curvature of the pattern arm. Therefore, due to the failure of the DPR/KLdiv criterion, decomposition and reconstruction do not achieve the removal of noise and redundant information. Although RPCA enhanced fault features, SDP patterns are still contaminated by residual noise and irrelevant information.

Using extensive experiments, it is found that meaningful SDP features can only be displayed when the noise content in the vibration signal is low and the fault features are sufficient. The intuitive expression is ① a clear signal waveform and ② the envelope spectrum can display the fault characteristics. We use the improved DPR/KLdiv criterion in this article to process the inner race fault signal in Figure 9c, and the analysis results are listed in Table 2.

The map of component selection drawn according to Table 2 is shown in Figure 12a. The threshold *adTH* can only exclude IMF1, similar to the analysis results in Figure 11a, and clearly cannot complete the feature extraction. The components selected using the improved threshold *aw* are IMF4~7, consistent with the analysis in Figure 10: not selecting IMF1~3 with noise and some low-frequency components unrelated to fault information (IMF8~14). The signal is reconstructed using the components selected by threshold *aw*, and the waveform and envelope spectrum of the signal are shown in Figure 12b,c. The envelope spectrum is similar to that in Figure 11c and can display fault information. The difference is that using the improved DPR/KLdiv criterion reduces noise pollution on the waveform. Based on the SDP pattern calculated from the reconstructed signal (Figure 12d), the changes in curvature and thickness of the pattern arm can be observed. The improved DPR/KLdiv criterion can accurately select fault feature components, and the decomposition and reconstruction method combining EEMD and improved DPR/KLdiv can accurately remove most of the noise and irrelevant information in the enhanced signal.

#### 3.1.3. Validation of the Feature Extraction Method

In order to verify the effectiveness of the FNHN-RPCA and decomposition and reconstruction SDP pattern extraction methods proposed in this article, they were compared with commonly used Wavelet Threshold Denoising and VMD optimized with the sparrow optimization algorithm. The content is as follows: ① the differences in curvature, thickness, and scattered and concentrated areas of points in SDP images of different fault types after feature extraction and ② the differentiation between different fault types of SDP images after feature extraction, where differentiation is compared and analyzed using the correlation between two different fault types of SDP images and the correlation between SDP images of the same fault type. The correlation coefficient *R*(*A*, *B*) of the two-dimensional matrix of two types of fault images is calculated as follows:(31)$$R(A,B)=\frac{\sum {}_{m}\sum {}_{n}(A_{mn}-\bar{A})(B_{mn}-\bar{B})}{\sqrt{\left[{\sum {}_{m}\sum {}_{n}(A_{mn}-\bar{A})}^{2}\right]\left[{\sum {}_{m}\sum {}_{n}(B_{mn}-\bar{B})}^{2}\right]}}$$
where *A* and *B* represent the two-dimensional matrices of any two types of SDP images, $\bar{A}$ and $\bar{B}$, respectively, represent the mean of the two arrays, and *m* and *n* represent the size of the two-dimensional matrices of the images.

The Wavelet Threshold Denoising method sets the wavelet basis function and decomposition level to “sym10” and 6, respectively. We use this method to denoise the data of six types of health states of bearing and convert them into SDP images (*H* = 2, *ζ*
=30°), as shown in Figure 13a. Among them, only the SDP image with outer race fault@3 has restored a small amount of meaningful information, still containing much noise. In order to avoid accidental sampling, ten samples with the same parameters and health status were randomly selected and binarized, and then we superposed these two-dimensional matrices into a representative sample. Due to the different sizes of the six representative sample matrices generated in MATLAB, it is necessary to extract a local matrix of the same size centered on the midpoint of the image matrix to reflect the differences between the six representative samples. The extracted representative samples are shown in Figure 13b. It can be clearly observed that there is no clear distinction between these representative samples, and the SDP patterns processed using wavelet thresholding are still contaminated by noise.

We calculated the correlation coefficients between any two representative samples and the average correlation coefficients of the ten samples comprising each representative sample, as shown in Table 3. Table 3 shows that the correlation coefficient between the representative samples of normal bearings and faulty bearings is between 0.7130 and 0.8649, while the average correlation coefficient between the ten samples that make up the representative samples of normal bearings is 0.6826. The similarity among the SDP pattern images of normal bearings is lower than the similarity between normal and faulty bearing images. At the same time, there is a phenomenon where the similarity between images of the same type is lower than that among images of different types in other fault types. It can be inferred that the differences among SDP patterns in different health states are relatively small, and the stability of these features is poor. Based on SDP features, the health status of bearings may be misjudged.

In the method using VMD optimized with the sparrow algorithm, the local maximum envelope kurtosis of the signal is used as the objective function of the sparrow optimization algorithm to calculate the optimal number of components and the quadratic penalty factor of VMD, and components are selected based on the envelope spectral kurtosis of each component. We use VMD to process data of six types of faults, and the SDP characteristics of the processed data are shown in Figure 14a. Although the curvature and thickness of the pattern arm vary among different types of SDP features, this difference is not sufficient to distinguish each healthy state of the bearing. From Figure 14b of the representative samples, it can be seen that the pattern arm thickness of each representative sample is thicker, and the curvature change is not significant. Upon careful observation, it can be observed that: the SDP characteristics of normal bearings and ball faults are similar, with inner race faults being similar to outer race faults@12, and outer race faults@6 being similar to outer race faults@3, which will result in misjudgment of the bearing’s health status.

The correlation coefficient values calculated from these samples are shown in Table 4. In Table 4, (1) the correlation coefficient between normal and ball failure is 0.8473; (2) the correlation coefficient between inner race fault and outer race fault@12 is 0.8542; and (3) the correlation coefficient between outer race fault@6 and outer race fault@3 is 0.8507. The above correlation coefficients are all relatively large, which also indicates a high similarity between features. Meanwhile, the average correlation coefficients of each fault type SDP feature are 0.6250, 0.7012, 0.6824, 0.6827, 0.6862, and 0.7084, all of which are smaller than the correlation coefficients in (1), (2), and (3). So, after using the sparrow algorithm to optimize VMD, there will be a problem of low similarity between similar SDP features, which increases the instability of features. In contrast, the SDP features of the data are still polluted by noise, and the health status of the bearings cannot be accurately judged according to the features.

We use the SDP feature extraction method based on FNHN-RPCA and the decomposition and reconstruction proposed in this article to calculate the SDP features of six bearing health status data types. The processed SDP features and representative samples are shown in Figure 15. Compared with Figure 13a and Figure 14a, the overall differentiation of normal, inner race fault, ball fault, and outer race fault in Figure 15a is better. However, the three features of outer race fault@12, outer race fault@6, and outer race fault@3 are similar. The representative samples also verify this conclusion (Figure 15b). The correlation coefficients calculated based on the representative samples and the average correlation coefficients between the ten sets of data that make up the representative samples are shown in Table 5.

Table 5 shows that ① the correlation coefficients between the six types of health status representative samples of bearings range from 0.4246 to 0.7336, all of which are less than their own average correlation coefficient values. ② The correlation coefficients between the three representative samples of outer race fault@12, outer race fault@6, and outer race fault@3 are 0.7336, 0.7140, and 0.7066, which are slightly lower than their average correlation coefficients of 0.7933, 0.7950, and 0.7745, indicating high similarity among the three features. ③ The average correlation coefficients between the ten samples of the same class that make up each representative sample are relatively high, ranging from 0.7745 to 0.8076, indicating good stability of the features. It is speculated that three types of outer race fault may be misjudged. However, the SDP features of normal, inner race fault, ball fault, and outer race fault have a relatively large differentiation, which can clearly distinguish the health status of the four types of bearings. At the same time, the SDP features extracted using the method proposed in this article have a high similarity between the same fault types, indicating that the extracted features have good stability. Therefore, the health status of rolling bearings can be accurately distinguished based on the differences among SDP features.

### 3.2. Case 2. Laboratory Rolling Bearing Data

#### 3.2.1. Experimental Instruments and Experimental Data

In order to verify the universality of the proposed feature extraction method, this paper also tested the method on an experimental bench dataset. The rolling bearing data measurement system is shown in Figure 16. Two accelerators are installed in the vertical direction of the bearing housing. The sampling time for each group is about 10 s, with a sampling frequency of 50 kHz and a motor rotation frequency of 30 Hz. The load is constant in the experiment, and there is at most one faulty bearing in each experiment. The test bearing is a deep groove ball bearing whose model is 6205. The fault area is randomly scratched by manual labor. There are four bearing fault modes: normal, ball fault, inner race fault, and outer race fault.

#### 3.2.2. Comparison and Verification of Feature Extraction Methods

We randomly select ten vibration datasets for each type of health status, each containing 50,000 data points. Based on these data, we calculate 10 SDP features of various health states, and these similar features are binarized and superimposed. The representative samples calculated from these data are shown in Figure 17. Although there are differences in the thickness and curvature of the pattern arms representing samples, these SDP features still contain noise that affects changes in thickness and curvature. Next, we analyze the differentiation and stability of SDP features after feature extraction by analyzing the correlation coefficients between representative samples and the composition of various representative samples.

The plots of correlation coefficients calculated for the representative samples are shown in Figure 18. The average correlation coefficient between the SDP features of normal bearings is 0.8452, which is much higher than the correlation coefficient between normal features and the other fault features. At this point, the bearing failure can be accurately judged using the similarity between the features. The correlation coefficient between the SDP features of the inner race fault is 0.5887, which is smaller than the correlation coefficient between the inner race fault, ball fault, and outer race fault features. It is impossible to judge whether the bearing is an inner race fault based on the SDP features. It can be inferred that due to the interference of noise or irrelevant information, there are random factors in the data, resulting in low similarity between SDP features of inner circle faults. In addition, the SDP features of ball faults and outer race faults also have the same problem. From the above analysis, it can be found that the SDP features calculated from the original data lack stability and will have low similarity to SDP features of the same fault type due to the influence of noise or irrelevant information. At this point, the bearing fault cannot be accurately determined according to the SDP characteristics.

The original laboratory data is now processed using wavelet threshold transform and VMD optimized by the sparrow algorithm. The correlation coefficient graph between representative samples calculated is shown in Figure 19. It can be observed that the SDP features processed with both methods have a situation where the average correlation coefficient between similar features is smaller than the correlation coefficient between different features. Therefore, neither of these methods can accurately distinguish the problem of fault types.

We utilize the SDP feature extraction method based on FNHN-RPCA and the decomposition and reconstruction proposed to extract features from the above data, and the representative samples are shown in Figure 20. The thickness and curvature of each healthy state sample pattern have significant changes, and there is less interference noise around the pattern. According to the correlation coefficient calculated from the representative samples in Figure 20, as shown in Figure 21, it can be seen that the average correlation coefficient between the SDP features of healthy bearings is 0.8275, which is much higher than the correlation coefficient between normal features and other fault features. At this point, the similarity among SDPs can be used to accurately judge whether the bearing has malfunctioned. Meanwhile, SDP features of the other fault types also have such advantages. Based on the above analysis, the method proposed in this article reduces the impact of noise or irrelevant information on the stability of SDP features. It increases the differences between various types of SDP features. At this point, the bearing failure can be accurately judged based on the SDP features of the bearing.

## 4. Conclusions

This article proposes an SDP feature extraction method based on FNHN-RPCA and decomposition and reconstruction. This method uses FNHN-RPCA to decompose the fault information matrix calculated from the bearing vibration signal into a low-rank information matrix and a sparse background noise matrix. In addition, it uses EEMD to decompose the one-dimensional vibration signal recovered from the low-rank information matrix into information components of different frequency bands and uses the improved DPR/KLdiv criterion to select the components. Finally, it reconstructs the selected components and uses SDP to calculate the symmetric dot pattern of the reconstructed signal. Our proposed framework, combined with FNHN-RPCA and EEMD, can fully separate the interference noise, redundant information, and fault features in the original vibration signal into different frequency bands. The improved DPR/KLdiv can also accurately select components based on energy and correlation.

For public datasets, this method increases the average correlation coefficient between similar SDP features from 0.6826~0.7916 to 0.7745~0.8076, which increases the stability of SDP features of the same fault type. At the same time, it reduces the correlation coefficient between different types of SDP features from 0.6527~0.8649 to 0.4296~0.7336, which increases the difference between SDP features of different bearing fault types. For the experimental datasets: due to the relatively small amount of data noise and the relatively stable SDP feature, the average correlation coefficient between similar SDP images before and after processing has little change. The correlation coefficient between different types of SDP features decreased from 0.5622~0.7360 to 0.4461~0.6204, significantly increasing the differences in SDP characteristics in different health states. The experimental verification results show that this method can effectively reduce the impact of noise and redundant information on SDP features.

In addition, if the theoretical fault frequencies of various bearing faults are different or too close, the method proposed in this article can effectively distinguish these faults. Although the method proposed in this article has certain limitations, for example, it must be carried out under fixed speed and single-fault type conditions, it can provide readers with some ideas for using FNHN-RPCA for non-correlation denoising or feature extraction of vibration signals, and the improved DPR/KLdiv criterion can be well used in other component selection tasks. At the same time, we found that this method can be applied to fault feature extraction of variable speed and multiple fault types, and further research is underway.

## Figures and Tables

**Figure 1 sensors-23-08509-f001:**
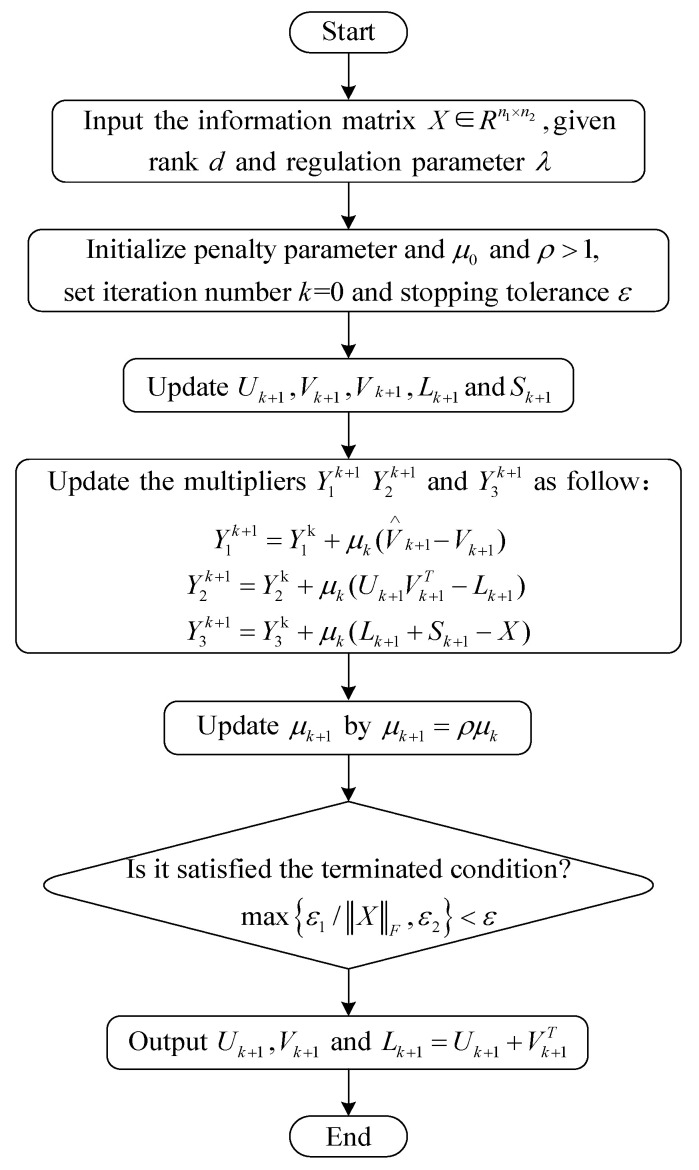
Flow chart of the ADMM algorithm solving Equation (10).

**Figure 2 sensors-23-08509-f002:**
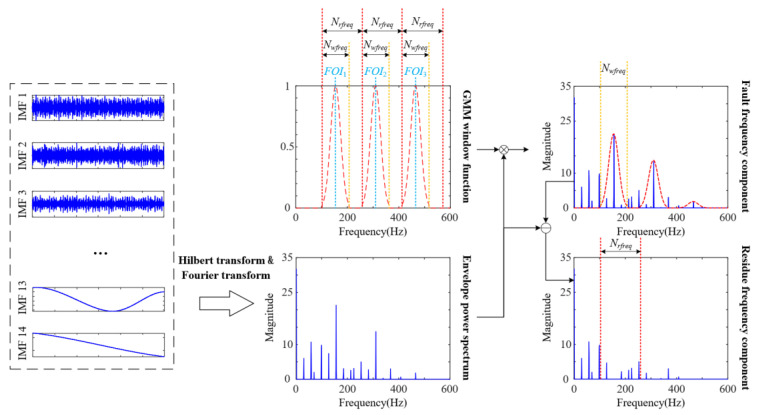
Flow chart of component decomposition based on GMM windows.

**Figure 3 sensors-23-08509-f003:**
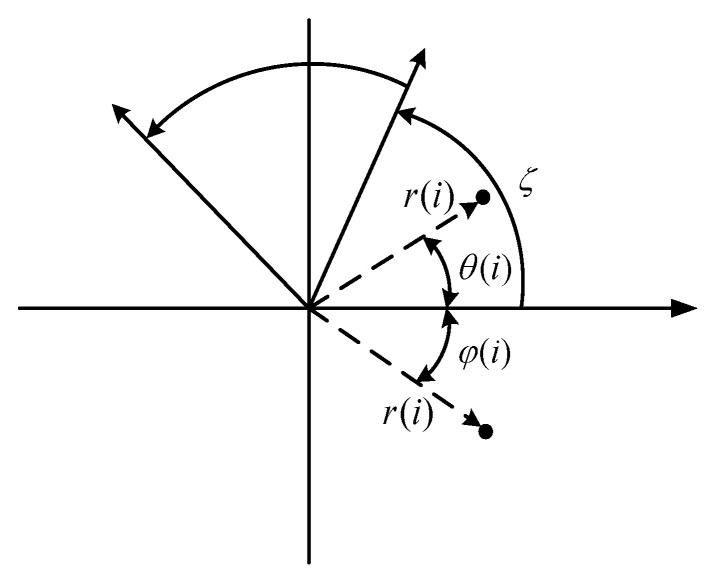
Principles of the SDP method.

**Figure 4 sensors-23-08509-f004:**
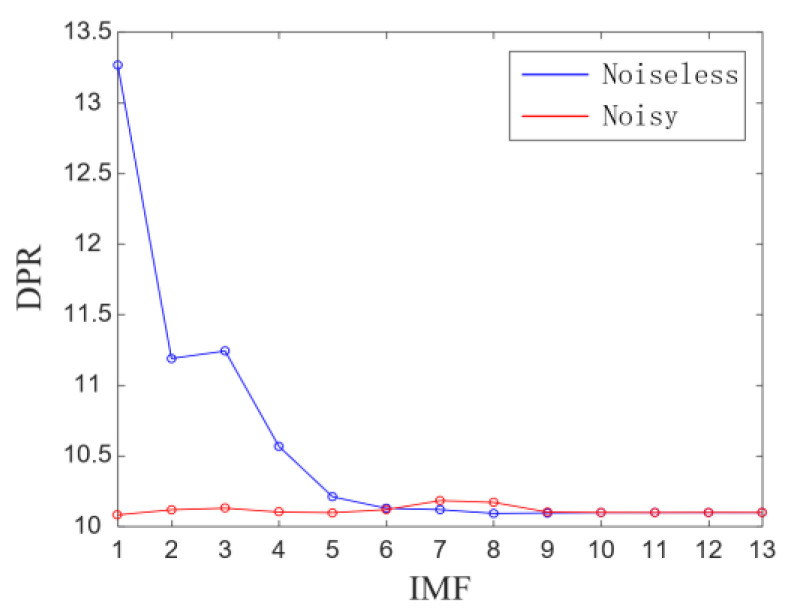
Comparison of DPR between noiseless and noisy components.

**Figure 5 sensors-23-08509-f005:**
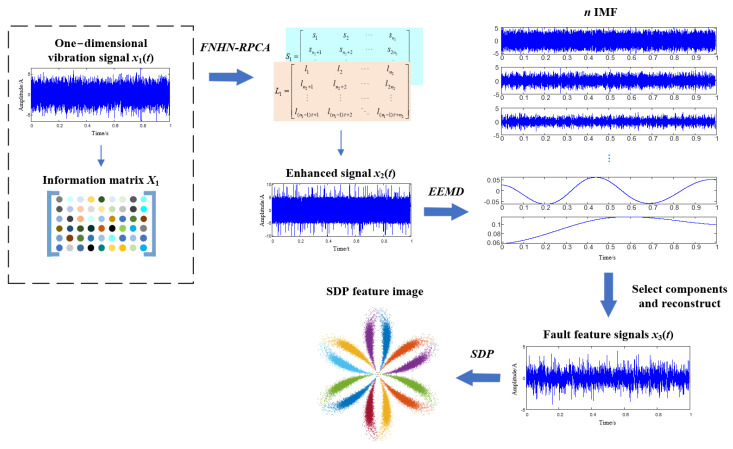
Flow chart of SDP feature extraction.

**Figure 6 sensors-23-08509-f006:**
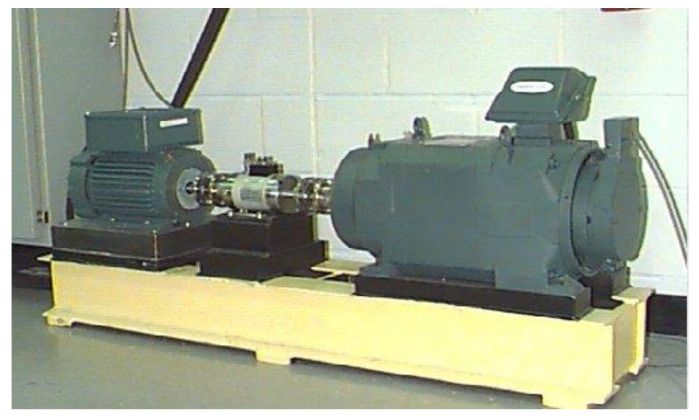
CWRU bearing test-rig.

**Figure 7 sensors-23-08509-f007:**
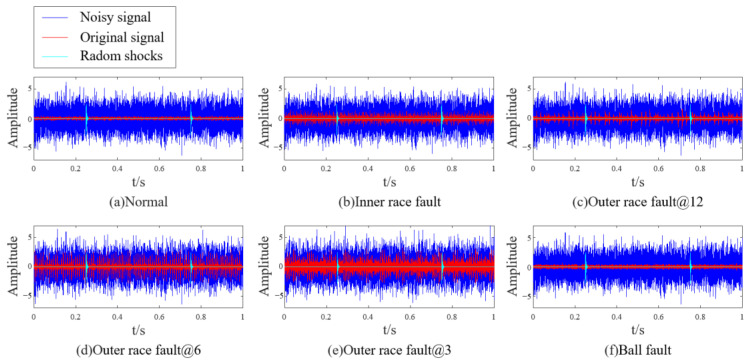
Noise data for six health states of bearings.

**Figure 8 sensors-23-08509-f008:**
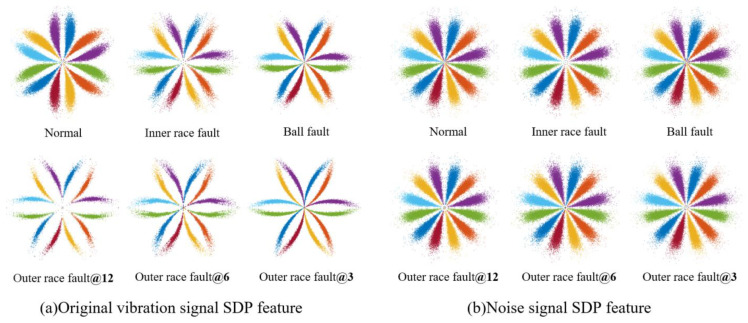
SDP pattern of the signals before and after noise addition for bearing class 6 health status.

**Figure 9 sensors-23-08509-f009:**
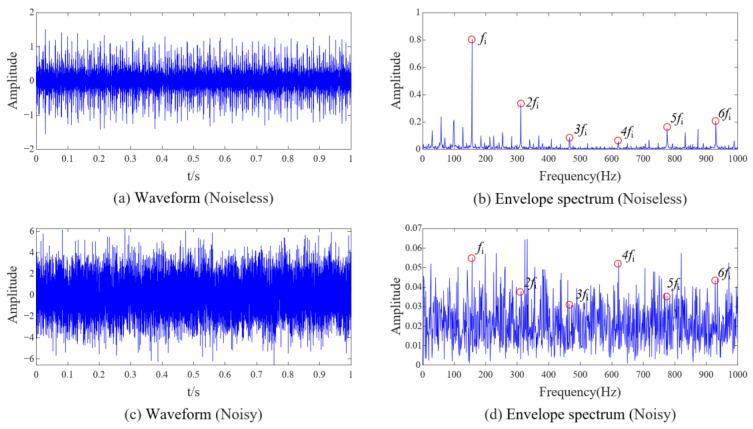
Comparison of waveform and envelope spectrum before and after data denoising.

**Figure 10 sensors-23-08509-f010:**
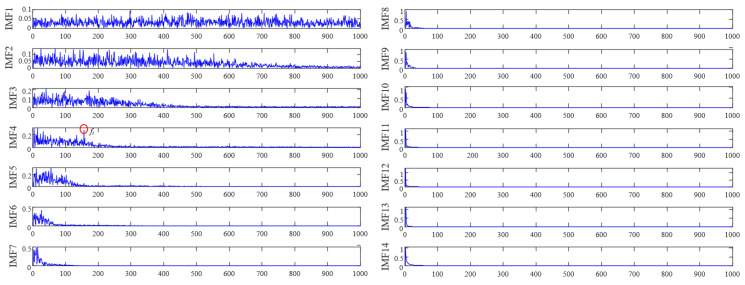
IMF envelope spectrum.

**Figure 11 sensors-23-08509-f011:**
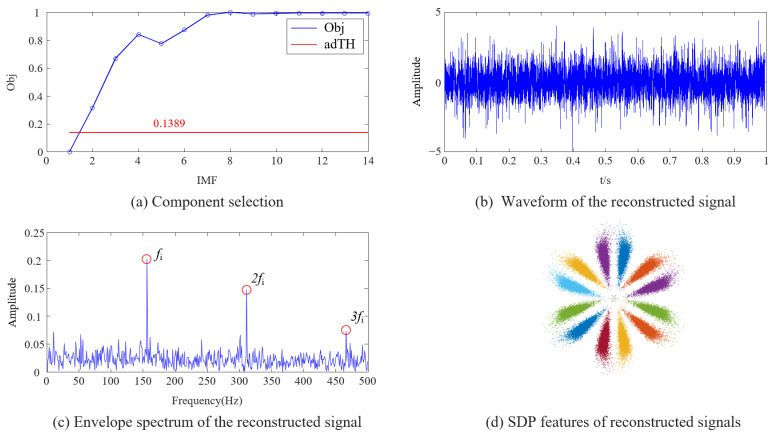
Analysis results of the original DPR/KLdiv criterion.

**Figure 12 sensors-23-08509-f012:**
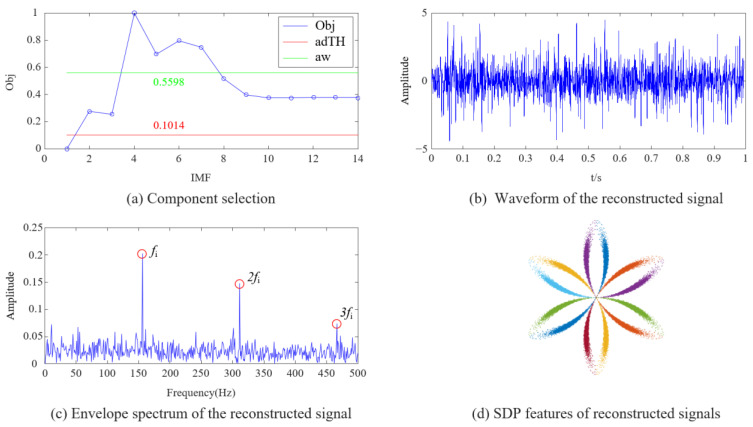
Analysis results of improved DPR/KLdiv criterion.

**Figure 13 sensors-23-08509-f013:**
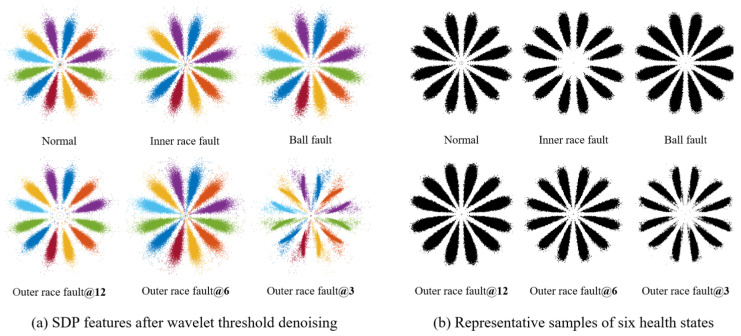
Analysis results of Wavelet Threshold Denoising.

**Figure 14 sensors-23-08509-f014:**
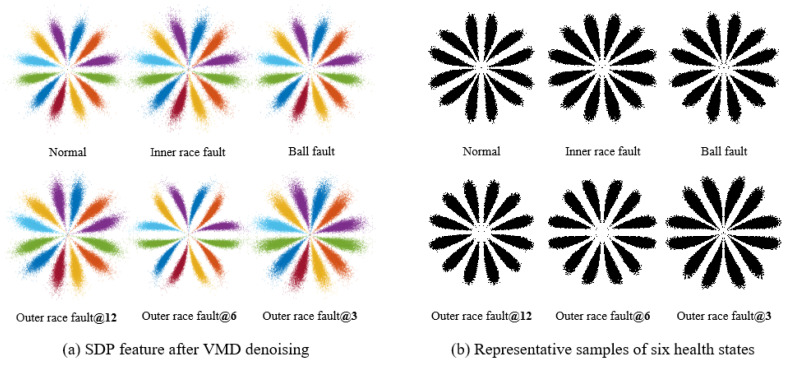
Analysis results of the optimized VMD.

**Figure 15 sensors-23-08509-f015:**
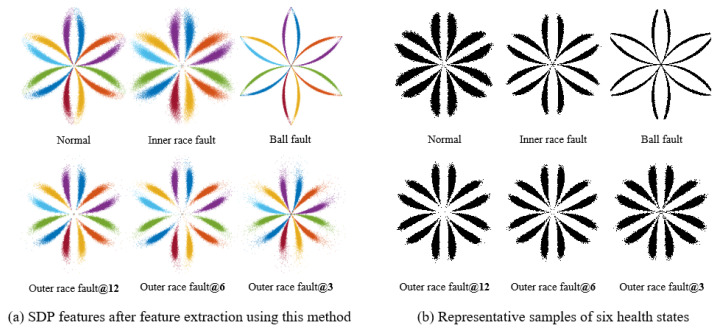
Analysis results of the method proposed in this paper.

**Figure 16 sensors-23-08509-f016:**
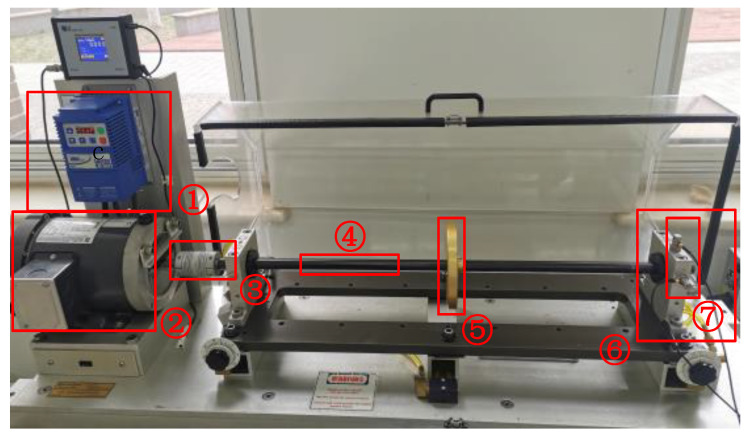
Rolling bearing test bench; ① motor driver, ② three-phase AC motor, ③ coupling, ④ rotating shaft, ⑤ rotating shaft, ⑥ fault bearing and its foundation, and ⑦ acceleration sensor.

**Figure 17 sensors-23-08509-f017:**
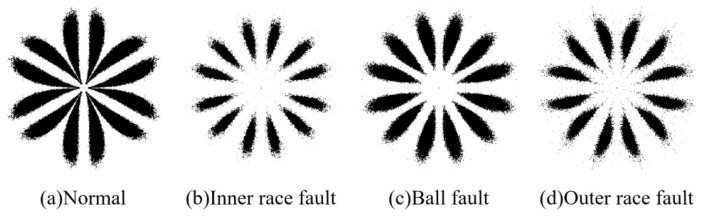
Representative samples of the original laboratory data.

**Figure 18 sensors-23-08509-f018:**
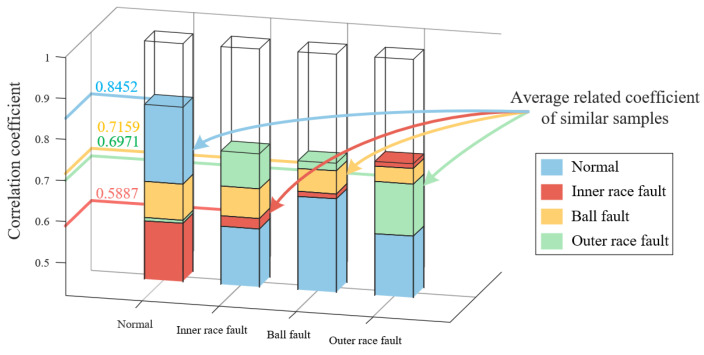
Correlation coefficient graph of representative samples of raw laboratory data.

**Figure 19 sensors-23-08509-f019:**
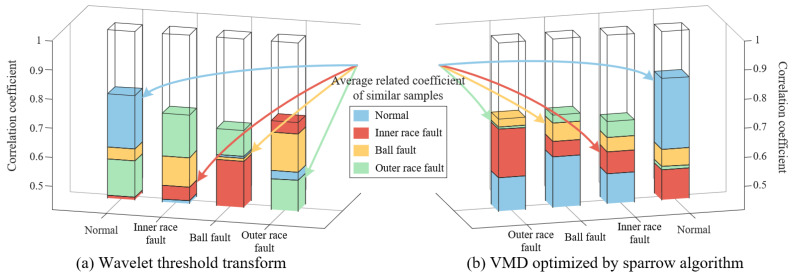
Analysis results of wavelet threshold transform and optimized VMD.

**Figure 20 sensors-23-08509-f020:**
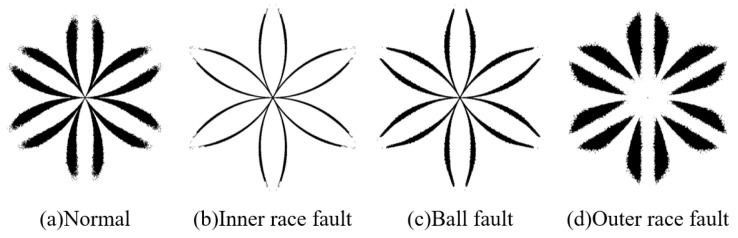
Representative samples of the feature extraction data.

**Figure 21 sensors-23-08509-f021:**
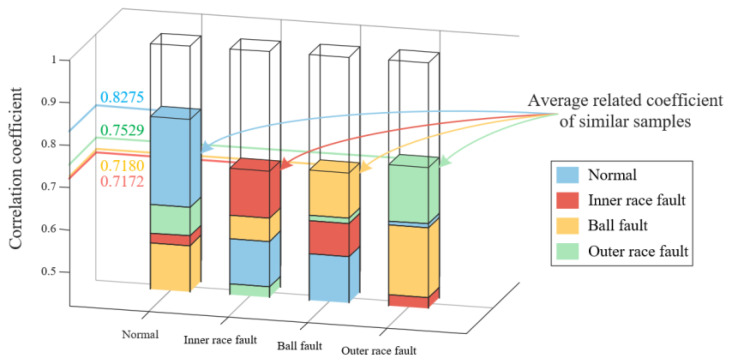
Analysis results of the proposed feature extraction method.

**Table 1 sensors-23-08509-t001:** Relevant parameter values of the DPR/KLdiv criteria.

IMF	1	2	3	4	5	6	7	8	9	10	11	12	13	14
DPR	10.1	10.1	10.1	10.4	10.2	10.2	10.2	10.1	10.1	10.1	10.1	10.1	10.1	10.1
KLdiv	1.13	0.99	0.88	0.86	0.86	0.84	0.81	0.8	0.8	0.8	0.8	0.8	0.8	0.8
Obj	0	0.32	0.67	0.84	0.78	0.87	0.98	1	0.99	0.99	0.99	0.99	0.99	0.99
adTH	0.1389													

**Table 2 sensors-23-08509-t002:** Relevant parameter values of the improved DPR/KLdiv criteria.

IMF	1	2	3	4	5	6	7	8	9	10	11	12	13	14
DPR	0.06	0.76	0.61	2.23	1.57	1.74	1.58	1.09	0.85	0.80	0.80	0.81	0.81	0.81
KLdiv	1.13	0.99	0.88	0.86	0.86	0.84	0.81	0.8	0.8	0.8	0.8	0.8	0.8	0.8
Obj	0	0.28	0.25	1	0.7	0.8	0.75	0.52	0.4	0.38	0.38	0.38	0.38	0.38
adTH	0.1014													
aw	0.5598													

**Table 3 sensors-23-08509-t003:** The correlation coefficient values of samples after wavelet threshold denoising.

	Normal	Inner Race	Ball	O-@12	O-@6	O-@3
Normal	1	0.7807	0.8649	0.8604	0.7781	0.7130
Inner race	0.7807	1	0.7663	0.7704	0.6527	0.6986
Ball	0.8649	0.7663	1	0.8791	0.7700	0.6868
O-@12	0.8604	0.7704	0.8791	1	0.7523	0.6717
O-@6	0.7781	0.6527	0.7700	0.7523	1	0.7563
O-@3	0.7130	0.6986	0.6868	0.6717	0.7563	1
Mean correlation coefficient	0.6826	0.6917	0.7680	0.7916	0.7502	0.6876

**Table 4 sensors-23-08509-t004:** The correlation coefficient values of samples after VMD.

	Normal	Inner Race	Ball	O-@12	O-@6	O-@3
Normal	1	0.7452	0.8473	0.7754	0.6200	0.5645
Inner race	0.7452	1	0.8285	0.8542	0.8057	0.7724
Ball	0.8473	0.8285	1	0.8314	0.7103	0.6537
O-@12	0.7754	0.8542	0.8314	1	0.7914	0.7464
O-@6	0.6200	0.8057	0.7103	0.7914	1	0.8507
O-@3	0.5645	0.7724	0.6537	0.7464	0.8507	1
Mean correlation coefficient	0.6250	0.7012	0.6824	0.6827	0.6862	0.7084

**Table 5 sensors-23-08509-t005:** The correlation coefficient values of the samples.

	Normal	Inner Race	Ball	O-@12	O-@6	O-@3
Normal	1	0.5145	0.4594	0.6114	0.5746	0.6144
Inner race	0.5145	1	0.4246	0.5758	0.5963	0.5410
Ball	0.4594	0.4246	1	0.4510	0.4703	0.4311
O-@12	0.6114	0.5758	0.4510	1	0.7336	0.7140
O-@6	0.5746	0.5963	0.4703	0.7336	1	0.7066
O-@3	0.6144	0.5410	0.4311	0.7140	0.7066	1
Mean correlation coefficient	0.7819	0.7952	0.8076	0.7933	0.7950	0.7745

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
