# Peer review of "A Symmetrized Dot Pattern Extraction Method Based on Frobenius and Nuclear Hybrid Norm Penalized Robust Principal Component Analysis and Decomposition and Reconstruction"

_sensors, 2023, doi:10.3390/s23208509_

Round 1

Reviewer 1 Report

This paper discusses a pattern extraction method. The manuscript brings new ideas. The text is well-written, and the manuscript is well-structured. The graphs are informative and have good quality. The manuscript is supported by a mathematical background and a proper literature review. There are some moments to be considered:

1.      The methodology can be more structured to make it clearer for readers.

2.      The abstract should be brief. Please try to avoid abbreviations.

3.      In literature review, try to avoid such [1-4] references. Try to explain them separately.

4.      Can this approach be used for other fault types e.g., cage damages?

5.      What is the main novelty of this study?

Reviewer 2 Report

Dear Authors,

I have some comments on your article:

1. Literature should be checked if there are no newer items. Especially from the last 18 months. The number of cited literature items should be significantly increased.

2. Please provide more details about the measurement system for data collection and about the types of bearings for which the data was analyzed

3. All indexes in symbols in text and equations should be checked carefully.

4. In the Conclusions section, please write something more about the real possibility of implementing the proposed methods in assessing the condition of rolling bearings.
